# Communication-Efficient and Privacy-Preserving Verifiable Aggregation for Federated Learning

**DOI:** 10.3390/e25081125

**Published:** 2023-07-27

**Authors:** Kaixin Peng, Xiaoying Shen, Le Gao, Baocang Wang, Yichao Lu

**Affiliations:** 1The Faculty of Intelligent Manufacturing, Wuyi University, Jiangmen 529020, China; kxinpeng@outlook.com (K.P.); lkiolyc@outlook.com (Y.L.); 2The State Key Laboratory of Integrated Service Networks, Xidian University, Xi’an 710071, China; bcwang@xidian.edu.cn; 3The Key Laboratory of Cryptography of Zhejiang Province, Hangzhou Normal University, Hangzhou 311121, China

**Keywords:** federated learning, privacy protection, verifiability, homomorphic hash function

## Abstract

Federated learning is a distributed machine learning framework, which allows users to save data locally for training without sharing data. Users send the trained local model to the server for aggregation. However, untrusted servers may infer users’ private information from the provided data and mistakenly execute aggregation protocols to forge aggregation results. In order to ensure the reliability of the federated learning scheme, we must protect the privacy of users’ information and ensure the integrity of the aggregation results. This paper proposes an effective secure aggregation verifiable federated learning scheme, which has both high communication efficiency and privacy protection function. The scheme encrypts the gradients with a single mask technology to securely aggregate gradients, thus ensuring that malicious servers cannot deduce users’ private information from the provided data. Then the masked gradients are hashed to verify the aggregation results. The experimental results show that our protocol is more suited for bandwidth-constraint and offline-users scenarios.

## 1. Introduction

Federated learning (FL) [1,2,3] is a distributed machine learning framework proposed by Google for privacy protection. In this framework, the involved users can conduct independent training and collaborate with the central server to obtain a high-performance shared global model. Unlike traditional distributed machine learning [4,5], users train their models locally instead of directly sharing private data. The server aggregates users’ trained local parameters or gradients rather than training on their data. Therefore, FL addresses concerns related to data privacy, data ownership, and data silos [6,7,8].

However, in the federated learning framework, the central server is generally considered untrusted and may infer the private data of users. Recent research [9,10,11] shows that adversaries can directly reconstruct users’ original data from the shared gradients. This means that an unreliable server can easily extract users’ private information from the transmitted gradients. Additionally, a malicious central server driven by self-interest may return incorrect results to users to conserve computing resources [12,13], leading to increased training iterations and a low-precision global model. Hence, protecting user privacy and ensuring data integrity are critical concerns within federated learning training protocols.

To address the above issues, Xu et al. [14] proposed a privacy-preserving and verifiable federated learning approach using secret sharing and homomorphic signatures. However, as the dimension of the vector increases, this method will greatly increase communication costs. The method proposed in reference [15] can efficiently validate results using commitment techniques and linear homomorphic hashing. However, it requires users to share hash and commitment values, resulting in additional communication overhead. Furthermore, ref. [16] introduced a model recovery attack on shared gradients hashes and suggested using shared vectors for verification. Nonetheless, the direct sharing of a verification vector can impose a significant communication burden on resource-constrained devices such as edge devices and sensors.

In this paper, we propose a communication-efficient and privacy-preserving verifiable aggregation federated learning protocol to facilitate training on limited bandwidth devices. Specifically, we utilize a single mask mechanism [17] to encrypt the gradients, ensuring privacy-preserving gradients aggregation. Additionally, we design a verification method to authenticate the integrity of the aggregated results against malicious server attacks. Our contributions can be summarized as follows:We propose a communication-efficient and privacy-preserving aggregation framework for the federated training process. By employing the additive homomorphism property of secret sharing, we protect the privacy of user information, while a homomorphic hash function enables verification.We devise a novel strategy to counter model recovery attacks. By employing homomorphic hashing on the masked gradients rather than directly hashing the gradients, we effectively protect users’ privacy against model recovery attacks on gradients hashes.We give a comprehensive security analysis to prove the high security of our scheme. Besides, extensive experiments demonstrate that our scheme has high communication efficiency with a moderate increase in computational costs.

The rest of the paper is organized as follows: we provide related work in Section 2, introduce preliminaries in Section 3, and describe the system model of our scheme in Section 4. After that, we discuss the construction of our scheme in detail in Section 5. The security analysis is provided in Section 6. Experiment evaluation of the proposed scheme is presented in Section 7. Finally, we conclude our paper in Section 8. Additionally, the abbreviations are used in the paper as follows.

## 2. Related Work

### 2.1. Privacy-Preserving FL

There have been many studies on federated learning about privacy protection. These studies primarily adopt the following three techniques: differential privacy (DP) [8,18,19,20], homomorphic encryption (HE) [9,21,22] and secure multiparty computing (SMC) [23,24,25,26,27,28].

In DP, Abadi et al. [18] proposed a training algorithm for differential privacy protection, which can obtain a more rigorous estimate of total privacy loss through quantitative analysis of privacy. Zhou et al. [19] proposed a differential privacy federated learning approach for detecting and filtering anomalous parameters uploaded from malicious terminal devices through edge node validation. However, methods based on differential privacy can cause a loss of global model accuracy owing to the introduction of random noise. In HE, Phong et al. [9] proposed a privacy-preserving deep learning model based on additive HE, which can realize high-precision model training in asynchronous federated learning. Park et al. [21] proposed a distributed homomorphic cryptosystem for implementing a privacy-preserving federated learning scheme. However, in the HE methods, the use of the same key pair by all users can lead to severe information leakage if users collude with the server. Additionally, the computation and transmission costs of ciphertexts are significant and may not be suitable for scenarios with resource-constrained devices that may dropout.

To address the privacy and dropout concerns on resource-constrained devices, some federated learning approaches have been proposed. Zhao et al. [29] characterized the optimal communication cost for the information-theoretic secure aggregation problem. Bonawitz et al. [23] designed a double-mask mechanism using SMC techniques for implementing privacy-preserving federated learning. To mitigate the supplementary overhead introduced by handling dropped users’ seeds in [23], So et al. [27] employed a one-shot aggregate-mask reconstruction approach to achieve secure aggregation in federated learning. Jahani-Nezhad et al. [30] proposed a secure aggregation protocol for federated learning systems that reduces communication overhead without compromising security. Schlegel et al. [28] proposed two privacy-preserving federated learning schemes to mitigate the impact of user dropout. Du et al. [22] proposed a threshold multi-key homomorphic encryption scheme to achieve secure, robust, and efficient federated learning training. Lu et al. [24] designed a secure aggregation protocol called Top-k SparseSecAgg to reduce communication cost and training time.

### 2.2. FL with Verifiable Computation

Many works have been proposed in verifiable federated learning. Xu et al. [14] proposed a privacy-preserving and verifiable federated learning scheme named VerifyNet. They used bilinear pairing and hash functions to achieve results verifiability, and used shamir secret sharing to achieve secure aggregation of gradients. However, the protocol requires users to use zero-knowledge proofs, and its communication cost increases linearly with the number of gradients. Furthermore, as mentioned in the attacks discussed in [16], the hashes allow malicious servers to infer users’ private data. Guo et al. [15] proposed VerifL, a communication-efficient and fast verifiable aggregation federated learning scheme, which achieved the verification communication independent of vector dimension. They utilized linearly homomorphic hash and commitment techniques to verify the aggregation results. However, it is vulnerable to the same attacks as mentioned in [16]. Zhang et al. [31] proposed a privacy-preserving and verifiable federated learning scheme that used Paillier encryption for gradients protection and bilinear aggregation signature for verification of the aggregation results. However, since users use the same encryption and decryption key, user privacy may be leaked if a malicious user colludes with the server. In the bilinear aggregation signature verification scheme, users generate signatures of gradients and send them to the server for aggregation. Users then verify whether the aggregated signature is equal to their own signature. However, a lazy server can aggregate only partial user gradients and signatures and send them to users for verification, which can pass the verification process. Additionally, the bilinear aggregation signature has a high verification cost.

Based on this, this paper aims to propose an aggregation verifiable federated learning method with efficient communication and privacy protection. Firstly, our approach can protect users’ private data against attacks from malicious servers and colluding users. On the other hand, in scenarios where bandwidth is extremely limited, our approach has a small communication overhead. Specifically, we compare the features of our approach with other existing solutions in Table 1.

## 3. Preliminaries

### 3.1. Federated Learning

Federated learning is a distributed machine learning framework. A server and multiple users work together to train a model. Suppose there are *n* users and a server in federated learning, and each user locally saves the dataset $D_{i}=\left\{\left(d_{ij},o_{ij}\right),j=1,2,\ldots ,J\right\}$.

Each user downloads the latest global model $w_{t-1}$ from the server during an iteration. The loss function $L_{f}^{i}\left(D_{i},w\right)=\frac{1}{\left|D_{i}\right|}\sum_{\left(d_{ij},o_{ij}\right)\in D_{i}}l\left(o_{ij};f\left(d_{ij},w\right)\right)$ is calculated by using the local dataset. $|D_{i}|$ represents the size of $D_{i}$. $f(d_{ij},w)$ denotes the predicted result and *w* denotes model parameters. $l(o_{ij};f\left(d_{ij},w\right))$ represents the loss result obtained by predicting the sample $\left(d_{ij},o_{ij}\right)$ on the given model parameters. To expedite model training, stochastic gradient descent (SGD) is utilized for computing the gradients $x_{i}=\nabla L_{f}^{i}\left(D_{i}^{k},w\right)$. $\nabla L_{f}^{i}$ represents the derivative of the loss function and $D_{i}^{k}$ represents a subset of the dataset $D_{i}$. Each user sends the calculated gradients $x_{i}$ to the server. The server calculates an updated global model, i.e., $w_{t}\leftarrow w_{t-1}-\alpha \sum_{i=1}^{n}\frac{|D_{i}|}{\left|D\right|}x_{i}$, where $\left|D\right|=\sum_{i=1}^{n}\left|D_{i}\right|$. The server then broadcasts the aggregated global model $w_{t}$ again, repeating the process until the global model converges or a certain number of iterations is reached.

### 3.2. Secret Sharing

Shamir secret sharing [32] divides a secret *s* into *n* secret shares, and only when the secret shares are greater than or equal to *t* can the secret *s* be reconstructed. For a Shamir (t,n) threshold scheme, let $U=\{u_{1},u_{2},\ldots ,u_{n}\}$ be a set of shared key users. In our scheme, $u_{i}$ represents the index number uniquely identifying the user. User $u_{s}$ represents a secret holder. The scheme is as follows:**SS.Share**(s,t,U): For the secret s∈GF(q), q>n, the user $u_{s}$ selects any t−1 positive integers to construct a polynomial $f\left(x\right)=s+a_{1}x+a_{2}x^{2}+\ldots +a_{t-1}x^{t-1}\left(\mathrm{mod}\;\;q\right)$. The user then calculates $f\left(u_{1}\right),f\left(u_{2}\right),\ldots ,f\left(u_{n}\right)$, and sends $\left\{u_{i},f\left(u_{i}\right)\right\}$ to user $u_{i}$.**SS.Recon**$\left({\left\{u_{i},f\left(u_{i}\right)\right\}}_{u_{i}\in U},t\right)$: Given any *t* of $\left\{u_{i},f\left(u_{i}\right)\right\}$, the secret holder can calculate the coefficients $s,a_{1},\ldots ,a_{t-1}$ of the polynomial f(x) using Lagrange interpolation and obtain the secret *s*. To reduce the computation, we adopt a simpler method of calculating the secret sum: $s=\sum_{j=0}^{t}f\left(u_{j}\right)\prod_{p=0,p\neq j}^{t}\frac{u_{p}}{u_{p}-u_{j}}.$

Shamir secret sharing satisfies additive homomorphism [33]. For example, suppose two users, $u_{1}$ and $u_{2}$, possess secrets $s_{1}$ and $s_{2}$, respectively. To secretly share the secret $s_{1}$ among *n* users in *U*, user $u_{1}$ employs the function **SS.Share** ($s_{1}$, *t*, *U*) to select a polynomial $f\left(x\right)=s_{1}+a_{1}x+a_{2}x^{2}+\ldots +a_{t-1}x^{t-1}\left(\mathrm{mod}\;\;q\right)$ and calculates *n* shares of $s_{1}$, denoted as $\left\{u_{i},f\left(u_{i}\right)\right\}$, which are sent to the corresponding users $u_{i}$. Similarly, user $u_{2}$ uses the function **SS.Share** ($s_{2}$, *t*, *U*) to select a polynomial $g\left(x\right)=s_{2}+b_{1}x+b_{2}x^{2}+\ldots +b_{t-1}x^{t-1}\left(\mathrm{mod}\;\;q\right)$, and distributes shares $\left\{u_{i},g\left(u_{i}\right)\right\}$ to the users $u_{i}$. Consequently, each user $u_{i}$ can locally compute $\left\{u_{i},\left(f\left(u_{i}\right)+g\left(u_{i}\right)\right)\right\}$. Then, in collaboration with other users where the number of users involved is greater than *t*, the secret $s_{1}+s_{2}$ can be reconstructed using the function **SS.Recon** (${\left\{u_{i},\left(f\left(u_{i}\right)+g\left(u_{i}\right)\right)\right\}}_{u_{i}\in U}$, *t*). Similarly, it can implement shared refactoring of any number of secrets.

### 3.3. Homomorphic Hash Function

A homomorphic hash function [34] includes three algorithms: **HHF.Gen**, **HHF.Hash** and **HHF.Eval**.

**HHF.Gen**$\left(1^{k},1^{d}\right)$: This algorithm initializes the system parameters by entering a random parameter *k* and a dimension *d*. The algorithm outputs common parameters pp, including the elliptic curve group *G* of order *q*, its generator *g* and *d* distinct elements $g_{1},g_{2},\ldots ,g_{m}\in G$.**HHF.Hash** (x): Input a *m* dimension vector *x*, the algorithm will output the hash value of $x:h\leftarrow \prod_{i=1}^{m}g_{i}^{x[i]}\in G$.**HHF.Eval**$\left(h_{1},h_{2},\ldots ,h_{k}\right)$: Input *k* hash values, this algorithm calculates the combination of hashes $h\leftarrow \prod_{i=1}^{k}h_{i}$. It satisfies the equation $h=\mathrm{HHF}.\mathrm{Hash}(x_{1}+x_{2}+\ldots +x_{k})$.

**Collision-resistant**: **HHF** is said to be collision-resistant if there is no adversary A that can satisfy the following experiment with non-negligible advantage.

${\mathrm{Adv}}_{A,\mathrm{HHF}}^{\mathrm{coll}\;}\left(k\right):=\mathrm{Pr}\left[\begin{array}{ccc} & & \mathrm{pp}\leftarrow HHF.Gen\left(1^{k},1^{d}\right)\\ HHF.Hash\left(x\right)=HHF.Hash\left(x^{\prime }\right) & : & \left(x,x^{\prime }\right)\leftarrow A\left(\mathrm{pp}\right) \end{array}\right]$. 

**One-way**: **HHF** is said to be one-way, if for any PPT adversary A that can satisfy the following experiment with non-negligible advantage for the security parameter *k* and the vector dimension m.

${\mathrm{Adv}}_{A,\mathrm{HHF}}^{\mathrm{one}\;}\left(k\right):=\mathrm{Pr}\left[\begin{array}{cc} & \mathrm{pp}\leftarrow HHF.Gen\left(1^{k},1^{d}\right)\\ x=x^{\prime }: & h\leftarrow HHF.Hash(x)\\ & x^{\prime }\leftarrow A\left(\mathrm{pp},h\right) \end{array}\right]$.

### 3.4. Key Agreement

In our scheme, Diffie-Hellman (DH) [35] key agreement protocols allow any two users to negotiate a key. This protocol consists of three algorithms: **KA.Setup**, **KA.Gen** and **KA.Agree**.

**KA.Setup**$\left(1^{d}\right)$: The algorithm inputs a security parameter *k* and outputs a public parameter pp.**KA.Gen**(pp): The algorithm will output a key pair $\left(pk_{i},sk_{i}\right)$ for user $u_{i}$ as input pp.**KA.Agree**$\left(sk_{i},pk_{j}\right)$: This algorithm inputs the private key $sk_{i}$ of user $u_{i}$ and the public key $pk_{j}$ of user $u_{j}$, and output an agreed key $s_{ij}$.

### 3.5. Authenticated Encryption

Authenticated encryption [36] includes three algorithms:**AE.Gen**(pp): The algorithm inputs a security parameter pp and outputs the secret symmetric key *k*.**AE.Enc**(k,m): The algorithm inputs a symmetric key *k* and a message *m*, outputs ciphertext *c*.**AE.Dec**(k,c): The algorithm inputs a symmetric key *k* and a ciphertext *c*, outputs plaintext *m* of *c*.

## 4. System Model

In this section, we first describe our system framework and communication model, introduce the threat model of our scheme, and define the design goals.

### 4.1. System Framework and Communication Model

We focus on how multiple users collaborate with the server to train a model with good performance. Each user is a mobile device that stores a dataset and can connect to the server. Our model consists of three entities: a trusted third party, a central server and users, as shown in Figure 1.

**Trusted third party (TA):** TA is mainly responsible for parameter initialization. It generates public and private key pairs and system parameters for each user participating in the training. Then it distributes these key pairs to the corresponding users and forward the general parameters.

**User:** Each user who stores the dataset locally can voluntarily choose whether to connect to the server and participate in model training. However, they may be due to network delay, system disconnection, system timeout, or active exit at any time [17]. In each iteration, users train with their local datasets to obtain the gradients and send the encrypted gradients and verification label to the server. After the completion of server-side aggregation, users are provided with the aggregated results and the corresponding verification proofs. Then, users verify the correctness of the aggregated results.

**Central server:** A server with sufficient computing resources which does not have a training dataset, and needs to collaborate with users to train a global model. The central server is responsible for aggregating gradients of encryption to update the worldwide model. It sends the aggregated results and verification proofs back to each user.

The protocol of our scheme is mainly divided into five rounds. Each interaction between the server and users is recorded as a round. If any round finds that the online users are less than *t*, the protocol stops immediately. The detailed content is as follows:Initialization: TA distributes system parameters to users and the public key of user will be shared.Round 1: User shares a secret with other online users, which will be encrypted using the public key.Round 2: User sends the encrypted model and the verification label to the server.Round 3: User produces a share which will be used for refactoring the secrets and sends it to the server. The server calculates aggregated results and aggregate verifiable labels.Round 4: User verifies the integrity of the aggregated results using verification proofs.

### 4.2. Threat Model

We adopted a threat model similar to reference [16], in which TA is a fully trusted entity that does not engage in any collusion. In this model, the server is considered malicious and attempts to deduce users’ training data. In addition, the server may manipulate the aggregation results to deceive users for unlawful purposes. We assume that users are honest-but-curious entities who correctly perform the training process and upload the masked gradients and verification label. However, users may try to deduce other users’ private data from the server’s returned results. Moreover, we consider the potential collusion between users and the server in practical federated learning scenarios, and the number of colluding users does not exceed the threshold value *t* in the context of secret-sharing. We exclude the scenario where users and the server collude to pass the verification process in our work. This is because multi-client verifiable computations fail to attain verifiability when the user colludes with the server, as demonstrated in [37].

### 4.3. Design Goals

In the system model, our work needs to meet the following two design objectives:**Privacy-preserving:** Privacy preservation is an essential part of federated learning. Therefore, our scheme must ensure that users’ private data can resist the adversaries’ attacks.**Integrity verification:** In federated learning, each user’s data are stored locally and trained locally. They can get a high-quality model by receiving the aggregated results returned by the server. If the server manipulates the aggregation results to deceive users, it could significantly impact their training. Therefore, our scheme ensures that the server cannot forge the aggregation results to deceive users into passing the verification process.

## 5. The Proposed Scheme

### 5.1. Overview

In this section, we provide a detailed description of the structure of our proposed solution. We will focus on addressing two issues: (1) how to securely aggregate without disclosing users’ sensitive information, (2) how to perform aggregation results’ integrity verification more efficiently in the context of limited bandwidth.

Before the system iteration training begins, the server will select a certain proportion of users to participate. The chosen users engage in five rounds of interactions with the server, as depicted in Figure 2. During these five rounds of interactions, users may exit the training protocol at any time due to network latency or other issues. We establish in Theorem 2 that even if some users prematurely terminate the training protocol, our proposed approach can still proceed without disruption. Each selected user possesses a unique and valid identity ID, denoted as *i* or *j*. Each user gets its local gradients after each round of training. Then each user adopts the single mask mechanism to encrypt gradients to obtain $\left[\left[x_{i}\right]\right]$ and hash function to calculate the verification label $T_{i}$ at the same time. Each user will upload $\left[\left[x_{i}\right]\right]$ and $T_{i}$ to the central server. After receiving the data uploaded by users, the central server aggregates the encrypted gradients and verification labels. Then it returns the aggregated results and verification proofs to users. Finally, each user will verify the correctness of the aggregation results. If the verification passes, the aggregation results will be used for the next round of local training. Otherwise, users will exit the training protocol.

### 5.2. Initialization

The initialization phase of the system is mainly guided by TA, which generates system parameters and public private key pairs for each user before training the model. First, TA generates public/secret keys $\left(u_{i}^{pk},u_{i}^{sk}\right)$ for each user and distributes them. In addition, a random seed *s* and hash parameters $g_{1},g_{2},\ldots ,g_{m}$ are also generated and forwarded to all users. Before users start training, the server randomly generates the learning rate α and initializes the model parameter *w* and sends them to users. Upon receiving these parameters, each user shares its public key with other users through the server to establish a shared encryption key for exchanging secrets.

### 5.3. Secure Aggregation

This section aims to protect users’ training model information from leakage. We use a single mask protocol to encrypt the gradients of users. Single mask mainly utilizes the homomorphism of secret sharing to achieve secure aggregation. The single mask method has stronger user dropout robustness than the double mask method.

We simply use a single mask to encrypt the gradients. Each user generates a mask $r_{i}$ and calculates $y_{i}=x_{i}+r_{i}$. The server calculates $z=\sum_{u_{i}\in U}y_{i}=\sum_{u_{i}\in U}x_{i}+\sum_{u_{i}\in U}r_{i}$. If the server wants to get the aggregated results $\sum_{u_{i}\in U}x_{i}$, it must know the aggregated mask $R=\sum_{u_{i}\in U}r_{i}$. However, the server cannot ask users to send $r_{i}$, as this would directly disclose the values $x_{i}$. A way of the server refactoring *R* is to utilize the homomorphism of secret sharing $R=Share_{\sum r_{i}}=\sum Share_{r_{i}}$, so the server only needs to get all the shares of $r_{i}$ to reconstruct *R*. User *i* generates *n* secret shares of $r_{i}$ before sending the encrypted gradients. One share of $r_{i}$ is represented by $r_{i}^{j}$, which represents the share of the secret $r_{i}$ sent by user *i* to user *j*. Thus, each user can obtain $r_{j}^{i}$ from other users. The $r_{j}^{i}$ is only part of the information held by the user *i* about $r_{j}$. The server will ask the online users *i* to send $R^{i}=\sum_{u_{j}\in U}r_{j}^{i}$, then the server calculates $\sum_{u_{j}\in U}R^{i}=R$. Since user *i* only knows $r_{j}^{i}$ but not $r_{j}$, and the server only knows $R_{i}$ but not $r_{i}$, they cannot learn anything about $x_{i}$. This single mask mechanism is much more robust against user dropout. If some users drop out of the protocol during the aggregation phase, then the server does not receive any model information for these users. The server only needs to send the set of online users during the aggregation phase to each user. Each user will then aggregate the $r_{j}^{i}$ value of the online users set. So long as the number of users online exceeds the threshold value *t*, the server can reconstruct *R*. So, the aggregation results can be calculated correctly.

As mentioned above, we have designed a simple protocol for secure summation on the server side. However, the effectiveness of our approach is wider than summation alone; it can also support weighted average aggregation. As shown in [38], a method is proposed to enhance models’ convergence speed and accuracy in a federated learning setting with non-i.i.d. data distribution by employing unbiased gradients aggregation. In this context, the server needs to aggregate $\sum_{i=1}^{n}\frac{|D_{i}|}{\left|D\right|}x_{i}$, where $\left|D\right|=\sum_{i=1}^{n}\left|D_{i}\right|$. To accommodate weighted average aggregation in this manner, we utilize two masks, $r_{i}$ and $r_{i}^{\prime }$ , to encrypt the users’ weighted gradients and data size. $r_{i}^{\prime }$ is shared among users in the same way as $r_{i}$. The encrypted data are represented as $y_{i}=\left|D_{i}\right|x_{i}+r_{i}$ and $c_{i}=\left|D_{i}\right|+r_{i}^{\prime }$. Afterwards, these encrypted data are sent to the server. The server can decode to obtain the sum without knowing the individual values of $|D_{i}|x_{i}$ and $|D_{i}|$, thus achieving effective and secure weighted aggregation.

### 5.4. Verification Phase

In this section, we will describe the verifiability of the protocol for the aggregated results. It is mentioned in [16] that if the value of the gradients hash is sent to the server as a verification label, the gradient entries generated by SGD rules form a bell-shaped distribution around zero because the values of the model parameters (gradients) are highly partial. In addition, since all users share the same private key, this is susceptible to brute-force attacks by malicious users in collusion with the server, so that the encoded gradients can be recovered in a brute-force attack on the victim’s homomorphic hash output in a relatively short time.

First, TA will send system parameters to users, including homomorphic hash parameters and a random number. The function of the random number *s* is to mask the gradients of the previous section twice, to prevent the server from forging the aggregation results and then taking the hash of the aggregation results as proof to deceive the user into passing the verification. User *i* encrypts the gradients as follows:(1)$$\left[\left[x_{i}\right]\right]=x_{i}+r_{i}+PRG\left(s\right),$$
where *PRG* is the pseudorandom generator. The user then utilizes the homomorphic hash function to generate a verification label: (2)$$T_{i}=\mathrm{HHF}.\mathrm{Hash}\left(x_{i}+r_{i}\right).$$

The user uploads $\left[\left[x_{i}\right]\right],T_{i}$ to the server.

Once the server receives encrypted gradients and verification labels, it performs two aggregations. The first aggregation is to calculate the aggregation results: (3)$$X=\sum_{u_{i}\in U}\left[\left[x_{i}\right]\right]-R=\sum_{u_{i}\in U}x_{i}+nPRG\left(s\right).$$

And the second aggregation is to calculate verification labels to generate the aggregation results’ proof as follows: (4)$$T=\prod_{u_{i}\in U}T_{i}=\mathrm{HHF}.\mathrm{Hash}\left(\sum_{u_{i}\in U}\left(x_{i}+r_{i}\right)\right).$$

Then the server returns the aggregation results *X* and the verification proofs T,R.

After receiving the aggregation results and verification proofs from the server, the user checks the following: (5)HHF.Hash(X−nPRG(s)+R)=T.

If the equation is true, it represents that the server returns the correct aggregation results, and the user will use the aggregation results for the next round of training. Otherwise, the verification fails, and the user exits the protocol.

### 5.5. Correctness of the Scheme

In order to verify the correctness of our scheme, we prove the following theorems.

**Theorem 1.** 
*If all users and the central server honestly execute our proposed protocol, the users can obtain the correct aggregation results, and the central server can pass the verification.*


**Proof.** The central server aggregates all the encryption gradients as follows:
(6)$$\begin{array}{ccc} \sum_{u_{i}\in U}\left[\left[x_{i}\right]\right] & =\sum_{u_{i}\in U}\left(x_{i}+r_{i}+PRG\left(s\right)\right)\; & =\sum_{u_{i}\in U}x_{i}+\sum_{u_{i}\in U}r_{i}+nPRG\left(s\right). \end{array}$$Next, the server calculates the aggregation results:
(7)$$X=\sum_{u_{i}\in U}\left[\left[x_{i}\right]\right]-R=\sum_{u_{i}\in U}x_{i}+nPRG\left(s\right).$$After receiving the aggregate results, the user calculates as follows:
(8)$$X-nPRG\left(s\right)=\sum_{u_{i}\in U}x_{i}.$$So, the user obtains the correct aggregation results and then performs the following hash calculation:
(9)$$H=\mathrm{HHF}.\mathrm{Hash}(\sum_{u_{i}\in U}x_{i}+R).$$If the server executes the protocol correctly, the aggregated hash is:
(10)$$\begin{array}{cc} T=\prod_{u_{i}\in U}T_{i} & =\mathrm{HHF}.\mathrm{Hash}\left(\sum_{u_{i}\in U}y_{i}\right)\;=\mathrm{HHF}.\mathrm{Hash}\left(\sum_{u_{i}\in U}\left(x_{i}+r_{i}\right)\right)\\ & =\mathrm{HHF}.\mathrm{Hash}\left(\sum_{u_{i}\in U}x_{i}+\sum_{u_{i}\in U}r_{i}\right)=\mathrm{HHF}.\mathrm{Hash}\left(\sum_{u_{i}\in U}x_{i}+R\right). \end{array}$$So, the server will pass the verification if Equation (Equation 9) equals Equation (Equation 10). □

**Theorem 2.** 
*Although some users may drop out of the training protocol, our proposed approach can still be effectively executed as long as the number of dropouts remains below the predefined threshold t.*


**Proof.** (Outline) In our proposed scheme, the impact of user dropout is limited to rounds 2 or 3. Let $U_{2}$ be the set of online users in round 2, from whom the server receives only the encryption gradients and verification labels. The server then performs an aggregation operation to eliminate $\sum_{u_{i}\in U_{2}}r_{i}$, and sends $U_{2}$ to the users in round 3. Users in round 3, who have saved the secret shares of $U_{2}$, send the sum of these user shares to the server. If the number of users in round 3 is greater than or equal to *t* (i.e., the server has received at least *t* shares of *R*), the server can reconstruct $R=\sum_{u_{i}\in U_{2}}r_{i}$. The verification process follows a similar approach as the aggregation process. □

**Theorem 3.** 
*The homomorphic hash function used in our approach can effectively verify the result of gradients aggregation.*


**Proof.** Each user computes a verification label, the hash of masked gradients $T_{i}=\prod_{k=1}^{m}g_{k}^{y_{i}\left[k\right]}$, where *m* is the gradients’ size. The server aggregates the verification labels as follows:
(11)$$\prod_{i=1}^{n}T_{i}=\prod_{i=1}^{n}\prod_{k=1}^{m}g_{k}^{y_{i}\left[k\right]}=\prod_{k=1}^{m}g_{k}^{\sum_{i=1}^{n}\left(x_{i}+r_{i}\right)\left[k\right]}=\prod_{k=1}^{m}g_{k}^{\left(X+R)[k\right]}.$$Based on homomorphic properties, we can derive the following:
(12)$$\prod_{i=1}^{n}T_{i}=\prod_{i=1}^{n}\mathrm{HHF}.\mathrm{Hash}\left(y_{i}\right)=\mathrm{HHF}.\mathrm{Hash}\left(y_{1}+y_{2}+...+y_{n}\right)=\mathrm{HHF}.\mathrm{Hash}\left(X+R\right).$$From this, we can conclude that the server can perform another form of aggregation on the $y_{i}$ without knowing its specific value, thus achieving effective verification of the system. □

## 6. Security Analysis

In this section, we demonstrate the security of our scheme in terms of users’ data privacy and integrity verification.

### 6.1. User Privacy

This section provides a demonstration that user privacy is protected. We use a similar proof method as that in [17]. We use Lemma 1 to prove that the encryption gradients of user uploads are secure.

**Theorem 4.** 
*Fix $n,s,U,r_{i}$ with |U|=n and $\{x_{i}{\}}_{u_{i}\in U}$, where $\forall u_{i}\in U,x_{i}\in Z_{R}^{m}$, there is :*

(13)
$$\begin{array}{c} \{\;r_{i}\in Z_{R}^{m}:x_{i}+r_{i}+PRG\left(s\right){\left(\mathrm{mod}\;B\right)}_{u_{i}\in U}\equiv I_{i}\in Z_{R}^{m}:I_{i}{\left(\mathrm{mod}\;B\right)}_{u_{i}\in U}\}, \end{array}$$


*where “≡” means having the same distribution, we have omitted this proof as it is simple.*


We use Theorem 4 to prove the privacy of the encrypted gradients uploaded by users. The encrypted gradients consist of $x_{i},r_{i}$ and *PRG(s)*. According to Theorem 4 , the distribution of $\left[\left[x_{i}\right]\right]$ is the same as $I_{i}$. If an adversary possesses the sum of $r_{i}$ and *PRG(s)*, he/she can obtain $x_{i}$ from $\left[\left[x_{i}\right]\right]$ due to the random distribution of $I_{i}$. However, the adversary must acquire at least *t* shares to reconstruct $r_{i}$ and obtain the sum of $r_{i}$ and *PRG(s)*. Thus, the adversary cannot obtain any information about $x_{i}$, indicating that our scheme maintains the security of the encrypted gradients during transmission, thereby ensuring user privacy. Since the encrypted gradients are not highly partial for the hash value uploaded by users, the encrypted gradients are hashed and then uploaded to the server. So, a malicious server cannot infer users’ private data according to the uploaded results, and it will not lead to the model recovery attacks like the one mentioned in [16].

### 6.2. Integrity Verification

**Theorem 5.** 
*According to the collision resistance and one way of **HHF**, a server cannot forge the aggregation results to pass the verification. The server returns the aggregation results X and the verification proofs T,R. Assume V=X+R; the result returned by the server is represented by V,T. We assume that the server is the adversary. There are only two ways for the server to falsify results, and we will show that falsifying proofs in either of these ways will not pass verification.*


(1) The server tries to forge *V* or *T* to pass the verification: In this case, the server forges *V* or *T* in round 3. We denote the forged results as $V_{f},T_{f}$ and the correct ones as V,T. Assume that the server only forges *V* to pass the verification, so $V\neq V_{f}$ and $T=T_{f}$. Let *H* denote the correct hash value computed by the user and $H_{f}$ represent the falsified hash value. We can obtain that from the third section:(14)$$H=T,H_{f}=T_{f}.$$

Since $T=T_{f}$, we can get $H=H_{f}$ according to Equation (Equation 14). However,
(15)$$\begin{array}{c} H=\mathrm{HHF}.\mathrm{Hash}(V-nPRG(s))\\ H_{f}=\mathrm{HHF}.\mathrm{Hash}\left(V_{f}-nPRG\left(s\right)\right). \end{array}$$

According to the collision resistance of **HHF**, we can conclude that $H\neq H_{f}$, since $V\neq V_{f}$. This contradicts our previous assumption. The same argument holds true if the server forges *T* instead. Therefore, the server cannot pass the verification by forging *V* or *T*.

(2) The server tries to forge *V* and *T* to pass the verification: In this case, since the server can pass the verification, the following equation that users calculate holds: (16)$$H_{f}=\mathrm{HHF}.\mathrm{Hash}\left(V_{f}-nPRG\left(s\right)\right)=T_{f}.$$

The adversary did not know *s* and $V\neq V_{f}$ because the adversary did not execute the aggregation protocol correctly. So,
(17)$$T_{f}=\mathrm{HHF}.\mathrm{Hash}\left(s^{\prime }\right),$$
where $s^{\prime }\in Z_{q}^{*}$ is a random vector guessed by the adversary. According to the one way of **HHF**, the probability of making Equation (Equation 16) valid is negligible. Therefore, the probability that the server wants to falsify the proofs to pass the verification is also negligible.

## 7. Model Analysis

### 7.1. Experimental Setup

We executed our experiment on a workstation with Java1.8 running Windows 11, equipped with an IntelCore i7-11700K 3.6GHz CPU and 16.0GB of RAM. We used elliptic curve Diffie-Hellman, standard Shamir’s *t*-out-of-*N* secret sharing and advanced encryption standard galois to implement **KA**, **SS** and **AE**, respectively.

### 7.2. Comparison with the Other Two Experiments Verifl [15] and Versa [16]

#### 7.2.1. Computation Overhead

In our scheme and the other two schemes, we compared the computational costs at different stages, as shown in Table 2. We omitted the comparison of the server overhead because the server is typically a powerful entity with negligible aggregation costs. Additionally, we omitted the consideration of round 0 as it does not incur any computation overhead. The underlined figures represent the verification costs. We fix the gradients’ size to 10,000.

The table shows that our aggregation cost is slightly lower than the other two approaches, while the verification cost is significantly higher. Our approach’s lower aggregation cost is attributed to the fact that we only use a single mask for aggregation. In contrast, the other two approaches employ double-mask encrypted gradients, requiring each user to compute the shared value ciphertext of two masks. The higher verification cost is due to adopting a more secure hash value to counter model recovery attacks from the adversary. In Versa, shared vector computation is used for verification labels. In VeriFL, each user employs linear homomorphic hashing and commitment for generating verification labels, which can lead to model recovery attacks, posing a significant privacy risk to users.

In the federated learning of privacy protection, user privacy, communication costs and the verifiability of the aggregated results are key issues during training, considering that modern edge mobile devices are typically equipped with high-performance processors but limited network bandwidth. Research [7,39] optimize the performance and efficiency of federated learning systems by increasing local computations and minimizing communication volumes. Furthermore, when it comes to transmitting sensitive data, a more cautious approach is required. It is essential to leverage local computational resources to minimize the transmission of private data. As seen in Section 7.2.2, the communication cost of our verification transmission is minimal, indicating that our approach is meant to reduce communication costs by increasing local computation overhead on edge mobile devices.

#### 7.2.2. Communication Overhead

We also compared the other two schemes in terms of the communication overhead for aggregation and verification.

Figure 3 depicts the aggregated communication cost between each user and the server. It demonstrates that our proposed approach, and the other two methods, increase linearly with the gradients’ size and the number of users in outgoing communication costs. Moreover, our protocol exhibits lower costs than the other two approaches. This discrepancy arises from using a single masking technique for aggregation in our scheme, whereas the other two employ double masking techniques. In the double-mask scheme, users must share two mask values to ensure the security of the aggregation.

Figure 4 illustrates the outgoing communication costs for verification between each user and the server. It shows that our proposed verification approach exhibits the lowest outgoing communication costs for both the user and the server, independent of the gradients’s size or the number of users. This advantage stems from transmitting only a single hash value for verification. However, in VeriFL, these two metrics are independent of the gradients’ size and increase linearly with the number of users. This is due to the necessity of users sharing their hash and commitment values. In contrast, these two metrics are independent of the number of users in Versa and increase linearly with the gradients’ size. In their approach, users must send a verification vector of the same dimensions as the gradients. Therefore, our verification method is particularly suitable for scenarios with multiple participating users and high-dimensional gradients.

### 7.3. Other Experiments of Our Scheme

This section analyzes our scheme’s computational and communication costs under different user dropout rates. In our experiment, we set the number of users n=500, and the gradients’ size m=10,000.

#### 7.3.1. Computation Overhead

From Figure 5a,c, it can be seen that the running time of the server decreases as the user dropout rate increases. The main reason is that the server only needs to aggregate gradients and tags and recover shares of online users. As shown in Figure 5b, the dropout rate does not affect the computational cost, as most of the time is spent on computing hashes, while the decryption time of ciphertext for online users can be ignored. Figure 5d indicates that the user’s computational cost of dropout is lower than that without dropout. This is because, in round 3, the user decrypts the online users’ ciphertext. Based on the experimental results, our scheme exhibits stronger user dropout robustness, as evidenced by a decrease in its running time with an increase in the dropout rate.

#### 7.3.2. Communication Overhead

Figure 6 shows the comparison of communication overhead and dropout rate between the server and each user. Specifically, Figure 6a and Figure 6c respectively demonstrate that the server’s communication overhead increases linearly with the number of users and gradients’ size and is independent of the dropout rates since the data sent by the server to each user remain constant regardless of the dropout rate. Additionally, Figure 6b,d show that the communication overhead per user under different dropout rates is uniform. Table 3 presents the outgoing communication overhead between the server and per user during different phases of our protocol. The underlined figures highlight the additional communication overhead associated with verification. Based on the table, we infer that the additional communication overhead caused by our verification process is minimal. This further confirms that our scheme is suitable for limited communication bandwidth settings.

## 8. Conclusions

This paper proposed a secure aggregation method that utilizes a single mask to encrypt gradients, providing a high level of privacy protection for users’ local gradients and allowing them to dropout. In addition, we devised a novel approach to verify the integrity of the aggregation results by hashing the masked value, capable of resisting the adversary’s attacks on the hash value in the federated learning framework. The experimental results demonstrate the efficiency of our approach in terms of communication costs for aggregation and verification. Therefore, our approach is effective in scenarios with limited bandwidth resources, solid computational capabilities, and frequent dropout of mobile devices.

However, our method also has some limitations. Firstly, the computational overhead of our scheme validation may be higher, which may not be suitable for devices with lower computing power, such as IoT devices and edge computing nodes. Secondly, our scheme requires five rounds of interaction to complete one iteration of training, leading to significant communication latency that hampers model training. Thirdly, in the event of incorrect results returned by the server, users cannot correct the aggregated results through available means. Therefore, it is necessary to further study the methods of reducing communication times and computing costs to achieve verifiable aggregation in federated learning training protocols.

## Figures and Tables

**Figure 1 entropy-25-01125-f001:**
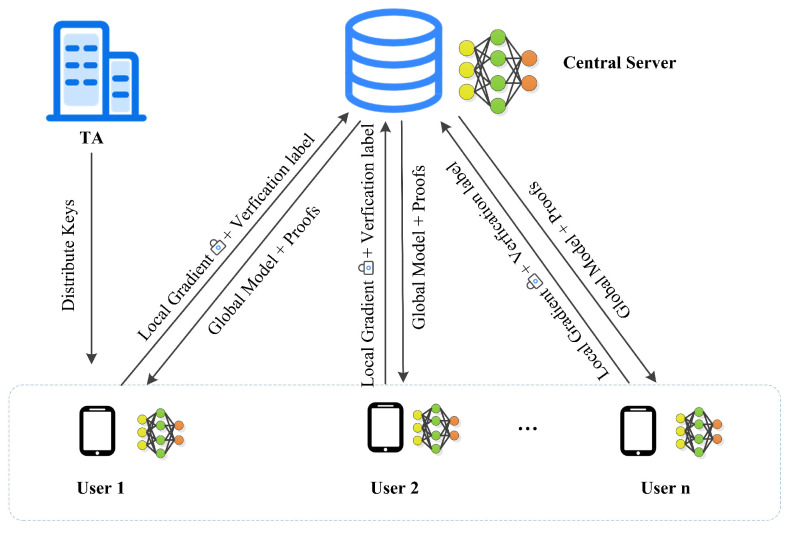
The system model of our scheme.

**Figure 2 entropy-25-01125-f002:**
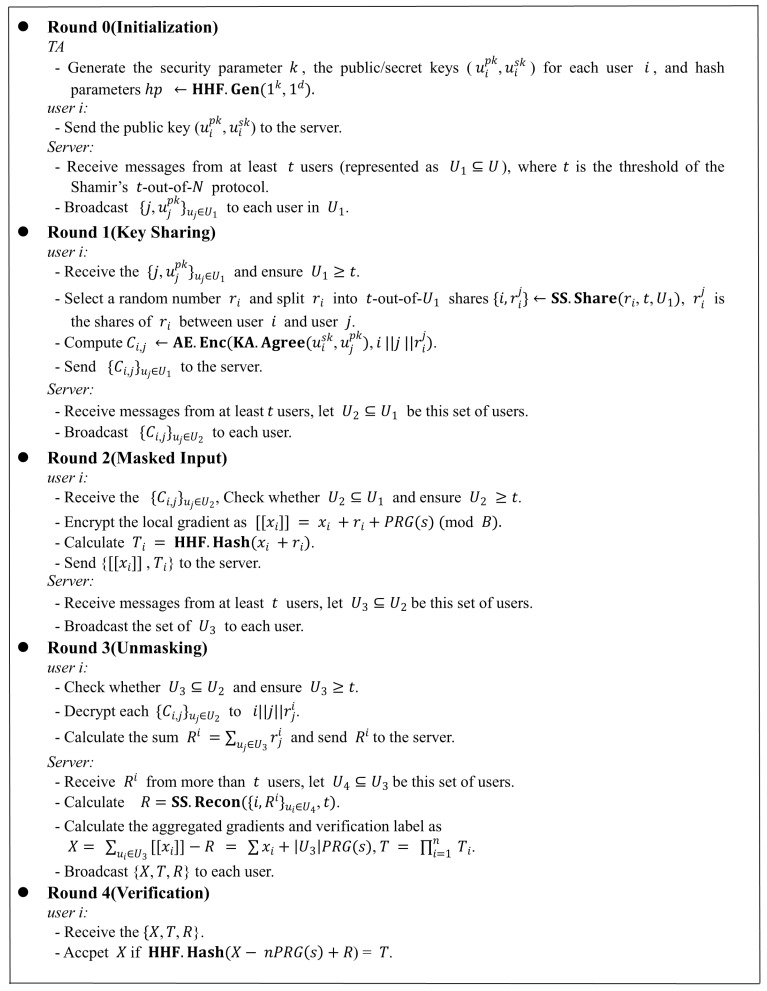
Our protocol in detail.

**Figure 3 entropy-25-01125-f003:**
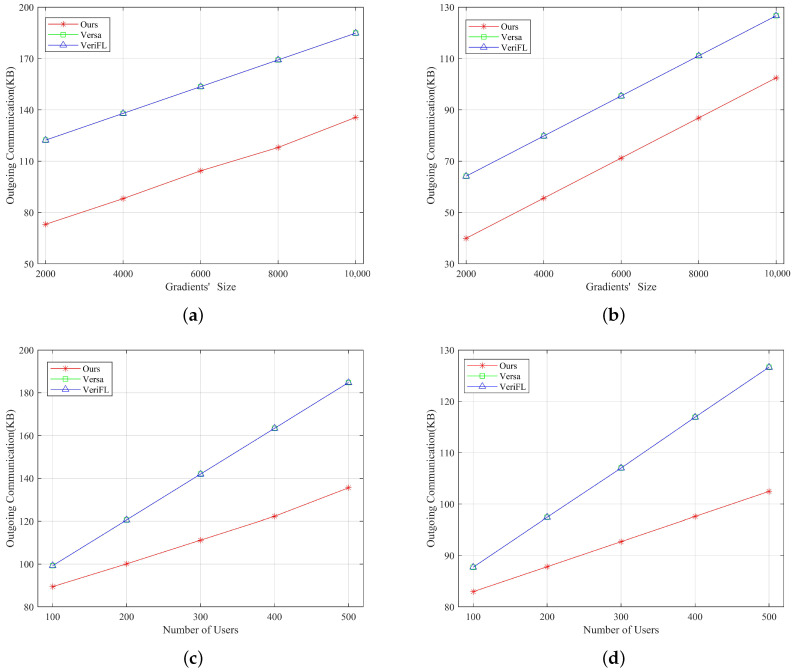
Comparison of outgoing communication overhead of our scheme with Versa and Verifl for aggregation. (**a**) Outgoing communication overhead of the server as the gradients’ size increases for aggregation. (**b**) Outgoing communication overhead per user as the gradients’ size increases for aggregation. (**c**) Outgoing communication overhead of the server compared to the different number of users for aggregation. (**d**) Outgoing communication overhead per user compared to the different number of users for aggregation.

**Figure 4 entropy-25-01125-f004:**
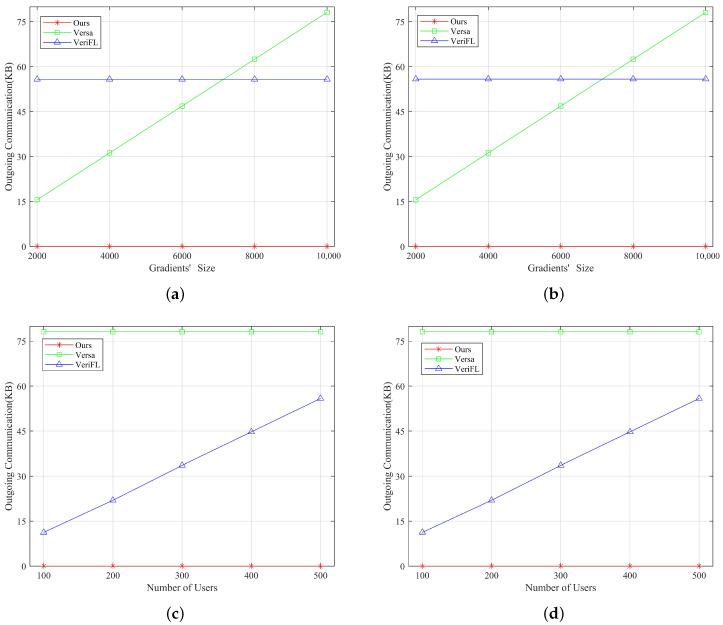
Comparison of outgoing communication overhead of our scheme with Versa and Verifl for verification. (**a**) Outgoing communication overhead of the server as the gradients’ size increases for verification. (**b**) Outgoing communication overhead per user as the gradients’ size increases for verification. (**c**) Outgoing communication overhead of the server compared to the different number of users for verification. (**d**) Outgoing communication overhead per user compared to the different number of users for verification.

**Figure 5 entropy-25-01125-f005:**
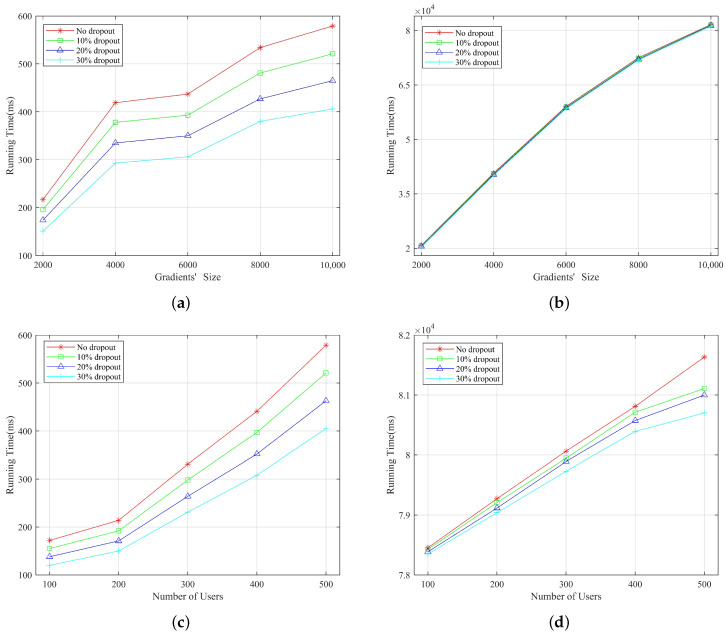
Thetotal computational overhead of the server and each user (**a**) computational overhead of the server versus different gradients’ size and dropout rates (**b**) computational overhead of each user versus different gradients’ size and dropout rates (**c**) computational overhead of the server versus different user numbers and dropout rates (**d**) computational overhead of each user versus different user numbers and dropout rates.

**Figure 6 entropy-25-01125-f006:**
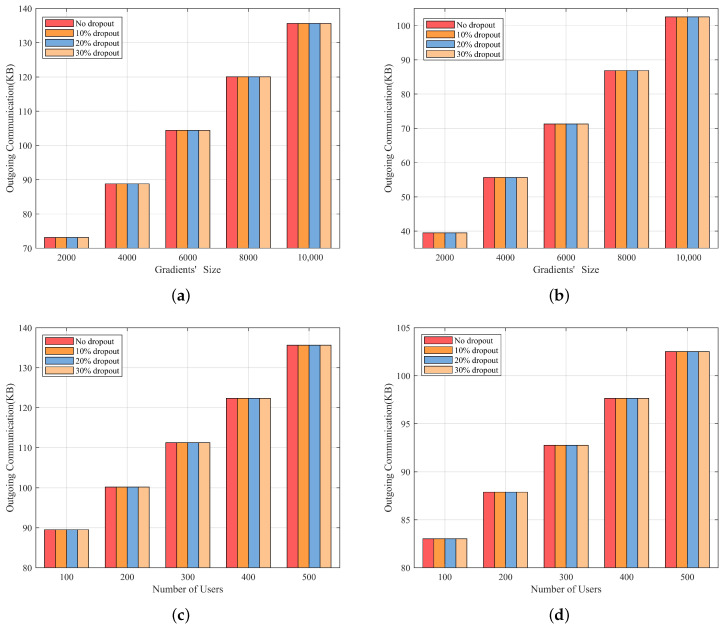
Total communication overhead between the server and each user (**a**) The communication overhead of the server versus different gradients’ size and dropout rates (**b**) the communication overhead per user versus different gradients’ size and dropout rates (**c**) The communication overhead of the server versus a different number of users and dropout rates (**d**) the communication overhead per user versus a different number of users and dropout rates.

**Table 1 entropy-25-01125-t001:** Functionality comparison.

Scheme	Privacy-Preserving	Verifiable	Drop-Tolerant	Resist Model Recovery Attacks	Communication-Efficient
PPML [23]	✓	-	✓	-	✕
PPDL [9]	✓	-	✕	-	✕
VerifyNet [14]	✓	✓	✓	✕	✕
VeriFL [15]	✓	✓	✓	✕	✓
Versa [16]	✓	✓	✓	✓	✕
Ours	✓	✓	✓	✓	✓

**Table 2 entropy-25-01125-t002:** Comparison of computation overhead with Versa and Verifl.

		Ours		Versa		Verifl	
		n=500	n=1000	n=500	n=1000	n=500	n=1000
**Aggregation phase**	1	2038 ms	4103 ms	2262 ms	4601 ms	781 + 7034 ms	1572 + 14,252 ms
	2	204 + 37,900 ms	210 + 37,903 ms	6123 + 6385 ms	6098 + 6465 ms	6236 ms	6248 ms
	3	2030 ms	4010 ms	2130 ms	4012 ms	2725 ms	5821 ms
**Verification phase**	4	37,913 ms	37,962 ms	120 ms	125 ms	2330 ms	4805 ms

**Table 3 entropy-25-01125-t003:** Outgoing communication overhead per user and server at different phases.

Num. User	Dropout	Entity	Aggregation Phase				Verification Phase
			0	1	2	3	4
500	0.00%	user	0.06 (KB)	24.23 (KB)	78.12 + 0.06 (KB)	0.03 (KB)	0 (KB)
		server	32.23 (KB)	24.23 (KB)	0.98 (KB)	78.12 + 0.07 (KB)	0 (KB)
	10.00%	user	0.06 (KB)	24.23 (KB)	78.12 + 0.06 (KB)	0.03 (KB)	0 (KB)
		server	32.23 (KB)	24.23 (KB)	0.88 (KB)	78.12 + 0.07 (KB)	0 (KB)
	30.00%	user	0.06 (KB)	24.23 (KB)	78.12 + 0.06 (KB)	0.03 (KB)	0 (KB)
		server	32.23 (KB)	24.23 (KB)	0.68 (KB)	78.12 + 0.07 (KB)	0 (KB)

## Data Availability

Not applicable.

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
