# Peer review of "Communication-Efficient and Privacy-Preserving Verifiable Aggregation for Federated Learning"

_entropy, 2023, doi:10.3390/e25081125_

Round 1

Reviewer 1 Report

This paper considers privacy preservation in a federated learning environment by using additive  homomorphic encryption of the gradients and using hashing to ensure that the server updates are not forged. The scheme uses well-known partial homomorphic encryption and secret sharing mechanisms but uses them in a somewhat interesting way, and thus does make a small contribution. However, there are many issues with the paper as it stands.

The threat model assumes that the server may manipulate aggregated gradients and the users are honest but curious. Yet, users are allowed to collude with the server.  An honest but curious user should not collude. It is not clear whether users can collude among themselves.

The basic description of FL is itself rather confusing. It seems to suggest that the gradients are simply averaged, as opposed to being weighted by the number of data points over which each client trains the model in each round. There is also confusion between notations for gradients and weights, with “g” having multiple meanings. The scheme is clearly designed for a simple summation, and there is no discussion how the scheme will work for more complex scenarios that may  weight the gradients in special ways to address issues if non-IID data, data quality, etc.. For example, the issue of bias in updates has been discussed in “Yao, Xin, et al. "Federated learning with unbiased gradient aggregation and controllable meta updating." arXiv preprint arXiv:1910.08234 (2019).”

The mechanism appears to make an implicit assumption that the same set of clients continue to participate. It does not consider how the scheme will operate if the number and IDs of the clients are chosen randomly. It has also not considered network or other issues that might cause a client to drop out at inconvenient times.

The results clearly show that the algorithm has much higher computation cost than the other two algorithms that it is compared with. However, this aspect seems to be discussed only rather briefly. The paper instead talks in more detail about the communications cost which are more favorable to the proposed algorithm. It is not clear what kind of environment is being considered in order to determine the importance of computation vs. communication overheads.  This is important to quantify the significance of reduced communication and substantially increased computation.   

The most objectionable thing about the paper is very poor English. For example, the introduction says “the central server is generally an infeasible entity that attempts to infer the user’s private data. As shown in [9], the central server can approach local data by numerical methods, thus causing user privacy disclosure.”

The English in introduction is entirely unacceptable and is adequate grounds for paper rejection unless improved substantially. What is a “violent attack”? This must be changed to something that makes sense.

Later sections have somewhat better English, but still full of incorrect grammar and sloppy writing, e.g., “Each user u_i calculate the …”.  “Assuming there are m(m >t) users ui0 , ui1 , . . . , uim−1 participates …”, “Then sent (ui, f (ui)) to user ..”, “user encryption to update the worldwide model”, “Server calculate aggregated results“, “u_s pick any t positive integer.”, “Then sent (ui, f (ui)) to user ui …”.  It appears that almost every other sentence has a typo or incorrect grammar. This level of language problems makes the paper entirely unacceptable as it stands.

Author Response

Dear reviewer:

   We are very grateful to your comments for the manuscript. According with your advice, we tried our best to amend the relevant part and made some changes in the manuscript. These changes will not influence the content and framework of the paper. All of your questions were answered below. And here we list the changes and marked in yellow in revised paper.

Q1: The threat model assumes that the server may manipulate aggregated gradients and the users are honest but curious. Yet, users are allowed to collude with the server.  An honest but curious user should not collude. It is not clear whether users can collude among themselves. 

R: We agree with you and apologize for not clarifying this aspect in our paper. In our proposed approach, we have considered the scenario where honest but curious entities may attempt to infer other users' data. We have also considered the possibility of collusion between users and the server during actual federated learning training. We have made the necessary modifications in the paper to address these points (Lines 237-242, page 6).

Q2: The basic description of FL is itself rather confusing. It seems to suggest that the gradients are simply averaged, as opposed to being weighted by the number of data points over which each client trains the model in each round.

R:Thank you for your comment and clarification regarding the basic description of federated learning (FL) in our paper. We agree with you and apologize for any misunderstanding caused by this description. In the revised version of the paper, We have modified this part of the content(Line 136, page 4).

Q3: There is also confusion between notations for gradients and weights, with “g” having multiple meanings.

R:We apologize for the repeated use of symbols g. We have addressed the issue of symbol repetition in our paper by replacing gradient w_i,g_i with "x_i" throughout the entire manuscript(Lines 133,136, page 4, lines 269,270,291, page 7, lines 313,316,318,332,335,337,342, page 8, lines 350,351,352,354,355,372,373, page 10, lines 383,386,387,388,390,399, page 11).

Q4:The scheme is clearly designed for a simple summation, and there is no discussion how the scheme will work for more complex scenarios that may  weight the gradients in special ways to address issues if non-IID data, data quality, etc..For example, the issue of bias in updates has been discussed in “Yao, Xin, et al. "Federated learning with unbiased gradient aggregation and controllable meta updating." arXiv preprint arXiv:1910.08234 (2019).”

R:We are grateful for the suggestion. We have thoroughly discussed the methodology of performing weighted averaging aggregation to accommodate more complex scenarios in Section 5.3 (Lines 308-319, page 8). This discussion will help readers understand the limitations and potential avenues for further research in extending the proposed scheme to handle these complexities.

Q5:The mechanism appears to make an implicit assumption that the same set of clients continue to participate. It does not consider how the scheme will operate if the number and IDs of the clients are chosen randomly.

R:Thank you for pointing out the assumption flaws in our paper. In our proposed approach, we have assumed that the selection of users for each iteration is random. We have revised the description of this mechanism in Section 5.1 (Lines 262-268, page 7) of the paper to accurately reflect this aspect.

Q6:It has also not considered network or other issues that might cause a client to drop out at inconvenient times.

R:We are grateful for raising the concern. In the revised version of the paper, we explicitly state that users can exit at any point during the 5-round interaction protocol we designed. In Theorem 2, we have provided proof demonstrating that user dropout does not impact the training process. We have made the necessary revisions to the relevant description in the paper (Lines 264-268, page 7).

Q7:The results clearly show that the algorithm has much higher computation cost than the other two algorithms that it is compared with. However, this aspect seems to be discussed only rather briefly. The paper instead talks in more detail about the communications cost which are more favorable to the proposed algorithm. It is not clear what kind of environment is being considered in order to determine the importance of computation vs. communication overheads.  This is important to quantify the significance of reduced communication and substantially increased computation.   

R:We sincerely appreciate your valuable feedback regarding quantifying the reduction in communication costs and the increase in computational costs. We apologize for the lack of detailed explanation in this section of our paper. We have considered this feedback and made revisions to the discussion on computational overhead. Besides, we have stated the environment and context being considered to determine the relative importance of these overheads(Lines 431-451, page 12).

Q8:The most objectionable thing about the paper is very poor English. For example, the introduction says “the central server is generally an infeasible entity that attempts to infer the user’s private data. As shown in [9], the central server can approach local data by numerical methods, thus causing user privacy disclosure.”

R:We apologize for the poor language of our manuscript. We worked on the manuscript for a long time and the repeated addition and removal of sentences obviously led to poor readability. We really hope that the language level have been substantially improved(Lines 24-32, page 1).

Q9: The English in introduction is entirely unacceptable and is adequate grounds for paper rejection unless improved substantially. What is a “violent attack”? This must be changed to something that makes sense.

R:We sincerely apologize for the poor quality of English in our introduction. We have taken this feedback seriously and made careful revisions to improve the clarity and coherence of the introduction section(Lines 18-34, page 1, lines 35-58, page 2). Furthermore, we have addressed the concern by replacing the term "violent attack" with "model recovery attack" throughout the paper (Lines 52, 54, page 2). We genuinely hope that the revised introduction meets the expected standards and contributes positively to the overall paper.

Q10:Later sections have somewhat better English, but still full of incorrect grammar and sloppy writing, e.g., “Each user u_i calculate the …”.  “Assuming there are m(m >t) users ui0 , ui1 , . . . , uim−1 participates …”, “Then sent (ui, f (ui)) to user ..”, “user encryption to update the worldwide model”, “Server calculate aggregated results“, “u_s pick any t positive integer.”, “Then sent (ui, f (ui)) to user ui …”.  It appears that almost every other sentence has a typo or incorrect grammar. This level of language problems makes the paper entirely unacceptable as it stands.

R:Thank you for bringing to our attention the extensive language and grammar issues in the later sections of the paper. We apologize for the poor writing quality and acknowledge that the current state of the paper is unacceptable in terms of language accuracy and clarity.

In light of your comment, we have carefully reviewed and made the necessary modifications to the paper (Lines 145-147, page 4) as suggested. We thoroughly revise and proofread the entire paper to address these language problems.

    We appreciate for your warm work earnestly, and hope that the correction will meet with approval. Should you have any questions, please contact us without hesitate. 

   Once again, thank you very much for your comments and suggestions.

Yours Sincerely,

Kaixin Peng.

Reviewer 2 Report

This paper introduces an effective secure aggregation verifiable federated learning system that is both communication-efficient and privacy-preserving. The authors utilize single mask technology to encrypt the gradients, allowing for privacy protection during gradient aggregation. This encryption ensures that malicious servers cannot deduce users' private data from the information they receive. To further guarantee the verifiability of the aggregation results, the authors employ hashes of the masked gradients. Through extensive experiments, the authors demonstrate that their scheme achieves communication efficiency with a moderate increase in computational cost. The results are interesting and novel. However, I highly recommend that the authors incorporate the following comments in the next review round:

  1. It appears that the authors may not have a full understanding of the concept of homomorphic secret sharing widely used in cryptographic literature. I suggest referring to the works of Elette Boyle, Niv Gilboa, Yuval Ishai, Huijia Lin, and Stefano Tessaro, such as "Breaking the circuit size barrier for secure computation under DDH" (CRYPTO 2016) and "Foundations of Homomorphic Secret Sharing" (ITCS 2018). Additionally, the paper at https://arxiv.org/abs/2111.10126 could provide valuable insights.

  2. Regarding homomorphic hash functions, for a rigorous proof of verifiability, authors should explicitly state the assumptions on which their construction relies. In cryptographic literature, verification properties are often denoted through the notion of security experiments. Please refer to https://ieeexplore.ieee.org/document/9833792 and related references for more information on this topic.

  3. It seems that homomorphic hashes can be replaced with any linearly homomorphic vector commitment, as indicated in Remark 9 of https://arxiv.org/abs/2301.11730 and Section 2.3 of https://arxiv.org/pdf/2111.12323.pdf. I recommend that the authors address this issue in the revised manuscript and consider constructions based on KZG commitments as well (see https://www.iacr.org/archive/asiacrypt2010/6477178/6477178.pdf).

  4. HHF abbreviation defined two times (see page 15)

"Adopting homomorphism secret sharing protects" on page 2 must be adopting homomorphic secret sharing and so on throughout the text. Please proofread it one more time. 

Author Response

Dear reviewer:

   We are very grateful to your comments for the manuscript. According with your advice, we tried our best to amend the relevant part and made some changes in the manuscript. These changes will not influence the content and framework of the paper. All of your questions were answered below. And here we list the changes and marked in yellow in revised paper.

Q1:It appears that the authors may not have a full understanding of the concept of homomorphic secret sharing widely used in cryptographic literature. I suggest referring to the works of Elette Boyle, Niv Gilboa, Yuval Ishai, Huijia Lin, and Stefano Tessaro, such as "Breaking the circuit size barrier for secure computation under DDH" (CRYPTO 2016) and "Foundations of Homomorphic Secret Sharing" (ITCS 2018). Additionally, the paper at https://arxiv.org/abs/2111.10126 could provide valuable insights.

R:We apologize for any inaccuracies or misunderstandings related to homomorphic secret sharing in the paper. We thoroughly review and study the works you have recommended. And we utilize the additive homomorphism property of secret sharing, not homomorphic secret sharing. We have made the necessary changes to the content regarding homomorphic secret sharing and replaced it with the term "additive homomorphism of secret sharing" throughout the paper to accurately reflect our approach(Line 49, page 2, lines 145-162, page 4).

Q2:Regarding homomorphic hash functions, for a rigorous proof of verifiability, authors should explicitly state the assumptions on which their construction relies. In cryptographic literature, verification properties are often denoted through the notion of security experiments. Please refer to (Multi-Server_Verifiable_Computation_of_Low-Degree_Polynomials)https://ieeexplore.ieee.org/document/9833792 and related references for more information on this topic.

R:Thank you for your valuable feedback and the suggested reference on verifiability. In the revised version of the paper, we have explicitly stated the assumptions on which our construction relies, particularly regarding the one-way and collision-resistant properties of the homomorphic hash functions used (Lines 174-180, page 5). Furthermore, in the security analysis, we have successfully demonstrated the integrity verifiaction of aggregated results based on these two assumptions(Lines 397-401, page 11).

Q3:It seems that homomorphic hashes can be replaced with any linearly homomorphic vector commitment, as indicated in Remark 9 of (Two-Server Private Information Retrieval with)https://arxiv.org/abs/2301.11730 and Section 2.3 of (Information Dispersal with Provable Retrievability for Rollups )https://arxiv.org/pdf/2111.12323.pdf. I recommend that the authors address this issue in the revised manuscript and consider constructions based on KZG commitments as well (Constant-Size Commitments to Polynomials and Their Applications)(see https://www.iacr.org/archive/asiacrypt2010/6477178/6477178.pdf).

R:Thank you very much for suggesting the use of linear homomorphic vector commitment based on KZG commitment as a means to achieve verifiability of aggregated results. It provided us with an interesting perspective. However, we regret to inform you that it is challenging for us to implement this suggestion due to the following reasons:

The linear vector commitment approach involves transforming a vector into a polynomial with the number of coefficients equal to the length of the vector. This allows us to convert the commitment of the vector into a commitment of the polynomial coefficients, providing linearity and irreversibility properties. These properties are well-suited to replace the homomorphic hash function. We have conducted some experiments to assess the computational cost involved. Firstly, the vector needs to be transformed into a polynomial using interpolation, which incurs a cost of O(m^3), where m is the length of the vector. Additionally, there are m exponentiations and m-1 multiplications required to compute the commitment, which has the same computational complexity as the homomorphic hash function we currently use. Furthermore, as the dimension of the vector increases, the overall computational complexity also increases. While we would like to further reduce the costs of interpolation and commitment, we have been unable to devise an efficient method to achieve this within our limited timeframe.

Switching from the current hash-based construction to linear vector commitment presents a significant challenge for us. Most of our experiments are based on the homomorphic hash function, which involves calculations and comparisons of computation and communication costs for different numbers of users and dimensions of vectors. Recalculating these costs for the comparative experiments would require significant time.

Once again, we appreciate your suggestion of linear vector commitment, which has provided us with a fresh perspective. In our future work, we intend to explore the application of linear vector commitment technology in verifiable federated learning. We also aim to focus on mitigating the overhead incurred by polynomial recovery.

   Q4:HHF abbreviation defined two times (see page 15)

   R: We apologize for this oversight and appreciate your keen observation. In the revised version of the paper, we have corrected this mistake in our paper(Line 530, page 15).

   Q5:"Adopting homomorphism secret sharing protects" on page 2 must be adopting homomorphic secret sharing and so on throughout the text. Please proofread it one more time.

   R:Thank you for identifying the error. Since we are not using homomorphic secret sharing, we have corrected“Adopting homomorphism secret sharing”to By employing additive homomorphism of secret sharing(Line 50, page 2) .

    We appreciate for your warm work earnestly, and hope that the correction will meet with approval. Should you have any questions, please contact us without hesitate. 

    Once again, thank you very much for your comments and suggestions.

Yours Sincerely,

Kaixin Peng.

Reviewer 3 Report

·      Line 32: choose a better word than “enemy”.

·      Citations are required to be sequentially numbered.

·      Authors are advised to review additional recent articles published in the similar areas where the privacy persevering approaches are presented for the resource constrained devices.

·      5.1 revise the heading.

·      Table 2 shows the dropout rate for only 500 users where the other researchers have compared the rates for 500 and 10000 users. Is there any reason for not evaluating the dropout rates for 1000 users.

·      Further the Table 2 and Table 3 does not clarify states whether the results are for aggregation or verification. It is advisable to show the results for both the aggregation and verification.

·      Similar to dropout, communication overhead results are not clear.

·      There are too many details on the general stuff and limited discussion to the key results.

·      The discussion on the proposed scheme with their limitations are required.

·      Conclusion is inadequate.

·      One of the major flaws in the study is the lack of justification for the effectiveness of the hashing to the gradient.

Author Response

Dear reviewer:

   We are very grateful to your comments for the manuscript. According with your advice, we tried our best to amend the relevant part and made some changes in the manuscript. These changes will not influence the content and framework of the paper. All of your questions were answered below. And here we list the changes and marked in yellow in revised paper.

Q1: Line 32: choose a better word than “enemy”.

R: Regarding your comment on Line 32, we have revised the sentence and replaced "enemy" with "adversary" to accurately describe the entity(Line 26, page 1).

Q2:Citations are required to be sequentially numbered.

R:We apologize for the use of a non-sequential citation format in our initial submission. In the revised version of the paper, we have made the necessary adjustments to ensure that all citations are sequentially numbered as required(Lines 20,23,25,29, page 1, lines 36,38,44, page 2). Thank you for bringing this to our attention.

Q3: Authors are advised to review additional recent articles published in the similar areas where the privacy persevering approaches are presented for the resource constrained devices.

R:Thank you for your suggestion. We agree that reviewing additional recent articles in the area of privacy-preserving approaches for resource-constrained devices would provide valuable insights and enhance the depth of our research. In the revised version of the paper, we have expanded our literature review to include a more comprehensive examination of recent publications specifically focusing on privacy preservation in resource-constrained devices(Lines 83-87, page 2, lines 88-94, page 3). This addition allows us to address the gap you mentioned and further strengthen the relevance and contribution of our work.

Q4:5.1 revise the heading

R:Thank you for your feedback regarding the heading in Section 5.1. In response to your comment, we have revised the heading in Section 5.1 to “Overview”(Line 257, page 7).

Q5:Table 2 shows the dropout rate for only 500 users where the other researchers have compared the rates for 500 and 10000 users. Is there any reason for not evaluating the dropout rates for 1000 users.

R:Thank you for bringing up this point regarding the evaluation of dropout rates for different numbers of users in Table 2. We appreciate your observation and the opportunity to clarify our revision. We have made some revisions to improve the logic and persuasiveness of our experiment. Taking into account the discussion on the impact of the dropout rate in Section 7.3 of our paper, we have concluded the computational overhead of calculating the dropout rate. Besides, compared to the verification overhead, it is not significant in comparison with the other two approaches. Therefore, we have removed the comparison of dropout rate in Table 2 and instead included a comparative experimental analysis of computational costs at different phases for a user population of 1000(Table 2, page 13). We agree that incorporating a comparative experiment with a user population of 1000 will provide a more robust comparison.

Q6:Further the Table 2 and Table 3 does not clarify states whether the results are for aggregation or verification. It is advisable to show the results for both the aggregation and verification.

R:Thank you for your suggestion regarding the clarification of results in Table 2 and Table 3. We agree with you that it would be beneficial to explicitly state whether the results presented in the tables are for aggregation or verification. In the revised version of the paper, We have added the costs of aggregation and verification phase to Tables 2 and 3, respectively, and highlighted the verification costs by underlining them(Page13, page 15). This clarification will provide readers with a clear understanding of the context and purpose of the reported metrics.

Q7:Similar to dropout, communication overhead results are not clear.

R:Thank you for your comment regarding the clarity of the communication overhead results in our paper. We have made revisions to the experimental section on communication costs in 7.2.2(Figure 3, page 13, Figure 4, page 14). Specifically, we have transformed the overall communication costs between each user and server into a comparative analysis of communication costs for aggregation and verification. We believe that this change will enhance the quality and comprehensibility of our research.

Q8:There are too many details on the general stuff and limited discussion to the key results.

R:Thank you for your feedback regarding the balance between general details and key results in our paper. We have rewritten seven chapters (Lines 426-456, page 12, lines 457-472, 478-480, page 13, lines 481-489, page 14) to simplify them, highlight the most important conclusions, and provide a more comprehensive analysis and interpretation of these results. We hope that this rewrite improves the description of the key conclusions.

Q9:The discussion on the proposed scheme with their limitations are required.

R:We agree with this comment. In the revised version of the paper, we have included a discussion on the limitations of our proposed approach in the conclusion section(Lines 509-517, page 15). These limitations encompass the relatively high computational overhead, the number of communication rounds, and the challenge of addressing incorrect results. By acknowledging these limitations, we aim to provide a comprehensive understanding of the drawbacks associated with our approach and potential areas for future improvement.

Q10:Conclusion is inadequate.

R:We apologize for the inappropriate description in our previous conclusion. We have revised the conclusion section to provide a more accurate and appropriate summary of our findings and limitations (Lines 500-503, page 14, lines 504-517, page 15).

Q11:One of the major flaws in the study is the lack of justification for the effectiveness of the hashing to the gradient.

R:We are grateful for the suggestion. In the revised version of the paper, we have provided a detailed justification for the effectiveness of the hashing technique applied to the masked gradient(Lines 368-373, page 10, lines 374-375, page 11).

    We appreciate for your warm work earnestly, and hope that the correction will meet with approval. Should you have any questions, please contact us without hesitate. 

    Once again, thank you very much for your comments and suggestions.

Yours Sincerely,

Kaixin Peng.

Round 2

Reviewer 2 Report

Authors have addressed all of my comments, and I recommend accepting the paper in its current form.

Author Response

Dear Reviewer,

   Thank you for reviewing our manuscript once again and acknowledging that we have made the some changes. 

    We hope that our revisions meet your expectations. If you have any further suggestions or comments regarding our modifications, please feel free to let us know. We are committed to further enhancing the paper to ensure its quality and accuracy.

     Once again, we sincerely appreciate your valuable feedback and suggestions for our research work. Thank you for your time and assistance.

Best regards,

Kaixin Peng.

Reviewer 3 Report

The suggested changes were made.

Author Response

(The authors gave the same response as above.)
